# Chiral acid-catalysed enantioselective C−H functionalization of toluene and its derivatives driven by visible light

Fuyuan Li[1,5], Dong Tian[1,5], Yifan Fan[1,2,5], Richmond Lee [3], Gang Lu[4], Yanli Yin[1], Baokun Qiao[1], Xiaowei Zhao[1], Ziwei Xiao[1] & Zhiyong Jiang [1,2]

Toluene and its derivatives are petroleum-derived raw materials produced from gasoline by catalytic reformation. These abundant chemical feedstocks are commonly used as solvents in organic synthesis. The C(sp$^3$)−H functionalization of these unactivated substrates has been widely used to directly introduce benzylic motifs into diverse molecules to furnish important compounds. Despite these advances, progress in asymmetric catalysis remains underdeveloped. Here, we report photoinduced radical-based enantioselective C(sp$^3$)−C(sp$^3$) coupling reactions of activated ketones with toluene and its derivatives by means of chiral acid catalysis. With a La(OTf)$_3$/pybox complex catalyst, a variety of chiral 3-hydroxy-3-benzyl-substituted 2-oxindoles, including many conventionally difficult-to-access variants, are obtained directly from isatins in high yields with good to excellent enantioselectivities. Acenaphthoquinone is also compatible with the use of a chiral phosphoric acid (CPA) catalyst, leading to another series of important enantioenriched tertiary alcohols.

[1] Key Laboratory of Natural Medicine and Immuno-Engineering of Henan Province, Henan University, 475004 Kaifeng, Henan, China. [2] Key Laboratory of Green Chemical Media and Reactions, Ministry of Education, School of Chemistry and Chemical Engineering, Henan Normal University, 453007 Xinxiang, Henan, China. [3] Singapore University of Technology and Design, Singapore 487372, Singapore. [4] School of Chemistry and Chemical Engineering, Shandong University, 250100 Jinan, Shandong, China. [5] These authors contributed equally: Fuyuan Li, Dong Tian, Yifan Fan. Correspondence and requests for materials should be addressed to Z.J. (email: chmjzy@henu.edu.cn)

The use of abundant industrial feedstocks to generate value-added products is always highly desirable in organic synthesis[1]. A classic paradigm is the direct C(sp3)−H functionalization of petroleum-derived toluene and its derivatives[2,3]. To enable the C−H activation of these substrates, which are generally used as solvents, many effective methods involving the oxidative insertion of transition-metal catalysts[4], hydrogen atom transfer (HAT)[5–7] and single-electron-transfer (SET) oxidation[8,9] have been developed. Nevertheless, asymmetric catalytic systems, which can serve to advance the development of chiral pharmaceuticals and chiral functional materials because of their ability to provide highly convenient and economical synthetic approaches, are still uncommon. The Davies group reported site-selective and enantioselective C–H functionalization of the secondary carbon[10] or primary carbon[11] in toluene derivatives by means of rhodium carbene-induced C–H insertion through using distinct chiral dirhodium catalysts (Fig. 1a). Recently, Melchiorre and co-workers skilfully took advantage of the high excited-state oxidation potential of iminium ion intermediates to perform SET with the aromatic rings of toluene and its electron-rich derivatives and furnish benzylic radical species (Fig. 1b)[12]. Covalent-bonding catalysis can be used to achieve good stereocontrol in the formation of C(sp3)–C(sp3) bonds. This elegant work further demonstrates that the use of a radical-based strategy offers improved regioselectivity with respect to transition-metal catalysed C−H activation procedures[13,14], especially in regard to the functionalization of toluene derivatives, by differentiating C(sp3)–H from C(sp2)–H due to the greater stability of a benzylic radical compared to that of an aryl one. The development of radical approaches with compatible asymmetric catalytic strategies is thus an extremely desirable goal.

In 1909, Paternò and Chieffi reported the synthesis of oxetanes by the photoinduced [2 + 2] cycloaddition of ketones with olefins[15]. This radical transformation was later named the Paternò–Büchi reaction and was proven to be triggered by the generation of triplet-state ketones under photoactivation[16,17]. The capability of photoexcited ketones to activate alkenes, alkynes and alkanes via an SET redox or HAT mechanism has been revealed in recent decades[18–24]. Accordingly, we wondered if a similar photochemical mechanism could serve to perform the direct C(sp3)−H functionalization of toluene and its derivatives, most likely furnishing benzyl-substituted tertiary alcohols. The use of an extrinsic chiral catalyst[25–32] to present efficient stereocontrol

might provide a straightforward approach to synthesize their optically pure variants, such as the abundant and biologically important chiral 3-hydroxy-3-benzyl-substituted 2-oxindoles, from commercially available isatins[33]. Although the enantioselective α-oxygenation of 3-benzyl-substituted 2-oxindoles could furnish these entities[34–41], the tedious preparation of the starting substrates and the limited types of benzyls, mainly owing to the expensive feedstocks or lack of viable synthetic methods, underscore the importance of exploring the feasibility of this challenging reaction.

Under irradiation with visible light, the triplet excited state of isatins generally shows n−π* reactivity and is a strong electron acceptor due to its rather high reduction potential ($E_{red\ 1/2}^*$)[23,24]. Our investigations also indicated that the introduction of distinct N-substituents can substantially alter the $E_{red\ 1/2}^*$ values of isatins (Supplementary Note 1). Given that the excited-state reduction potential of the N-acetyl isatin 1d (+2.06 V vs. SCE in CH3CN) is the same as that of [Acr+-Mes]ClO4 ($E_{red\ 1/2}^*$ = + 2.06 V vs. SCE in CH3CN), which as a photoredox catalyst allows the slow SET oxidation of toluene ($E_{ox}$ = +2.28 V vs. SCE in CH3CN)[9], this system is likely able to practically oxidize toluene, furnishing benzylic radical II, radical anion III[42] and a proton (Fig. 2a). Given the strong basicity of the oxygen anion of III, it might capture the chiral Brønsted acid[27,30,32] or Lewis acid[43] catalyst to form dioxindolyl or the dioxindolyl-like radical IV, which is a merostabilized carbon radical due to the capto-dative substitution of the radical centre by the electron-donating oxygen atom and the electron-withdrawing amide group[44]. According to the persistent radical effect[45], the coupling of radical IV with the transient benzylic radical II is viable and thus furnishes oxindoles, possibly in an enantioselective manner. Mechanistically, an HAT process[19] involving these photoexcited isatin species and toluene is also thermodynamically favourable owing to their high triplet-state energy ($E_T$, Fig. 2a and Supplementary Note 1), which should be enough to overcome the gap between the bond-dissociation energies (BDEs) of the O-H bond of the generated dioxindolyl radical[44] V (48.1–49.8 kcal × mol−1, Supplementary Note 1) and the sp3 C−H bond of toluene (88.5 ± 1.5 kcal × mol−1)[46]. The chiral acid catalyst also potentially provides stereocontrol for the radical coupling by interacting with the merostabilized dioxindolyl radical V to form the more electrophilic radical VI[31,43]. It is worth mentioning that neutral radicals II[47] and IV/V[48] readily undergo homocoupling, which can affect the achievement of a satisfactory yield. Meanwhile, the very low activation barrier of

**Fig. 1** Prior works. **a** Dirhodium tetraprolinate-catalysed reactions of aryldiazoacetates with toluene and its derivatives. **b** Photocatalytic C−H functionalization of toluene and derivatives enabled by excited-state iminium ion catalysis

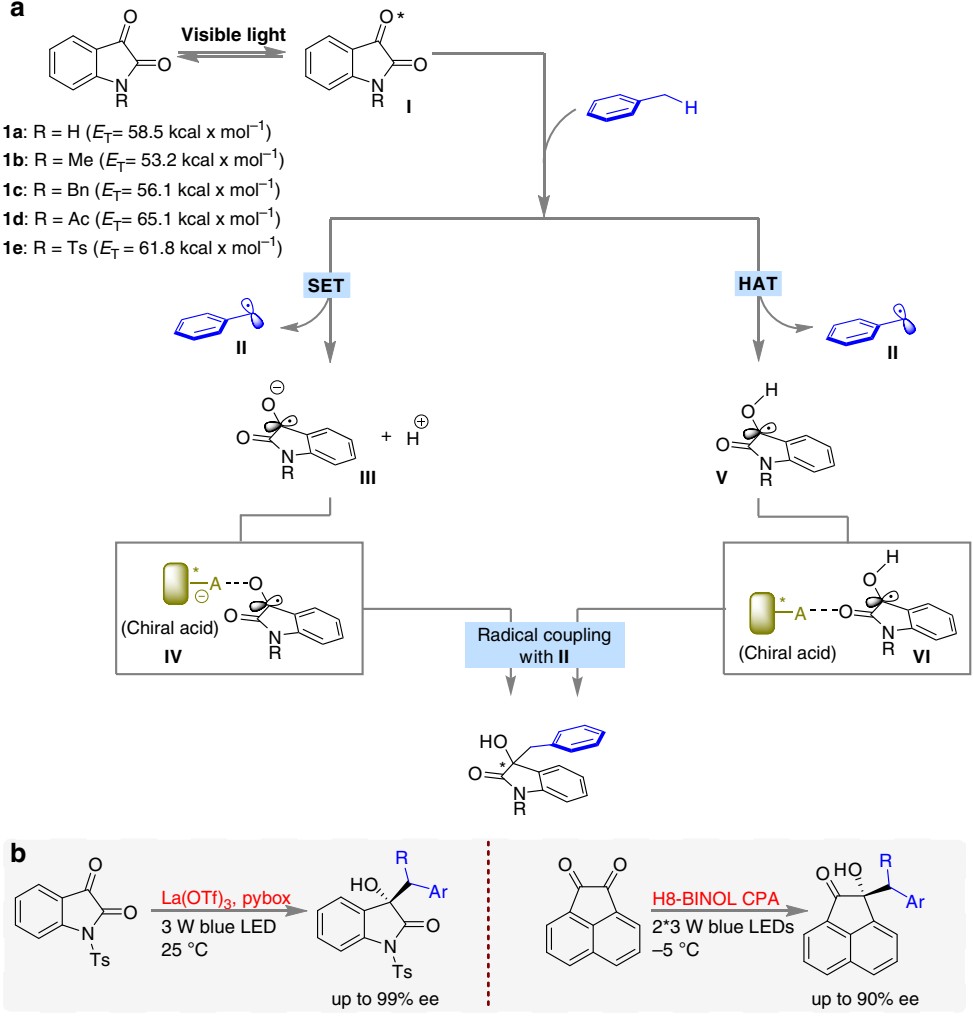

**Fig. 2** Outline of this work. **a** Design plan. **b** Chiral acid-catalysed enantioselective C−H functionalization of toluene and its derivatives by activated ketones

the radical coupling and competitive racemic background process constitute two crucial challenges for precise and absolute stereocontrol. Herein, we report the investigation and realization of this hypothetical scenario (Fig. 2b). A wide array of valuable and challenging-to-synthesize chiral tertiary alcohols were expediently prepared by the reaction of toluene and its electron-rich and electron-deficient derivatives with isatins and acenaphthoquinone by using chiral Lewis acid or Brønsted acid catalysis.

## Results

**Reaction optimization.** The viability of the plan was first examined by the transformations of isatins **1a-e** with toluene as the solvent and irradiation with a 3 W blue LED (entries 1−5 in Supplementary Table 1). Isatin **1a** was found to undergo poor chemical conversion, and the reaction delivered a complex crude mixture. Furthermore, no reaction was observed for **1b** or **1c**. When using **1d** and **1e**, to our delight, the corresponding products, *rac*−**2d** and *rac*−**2e**, were obtained in 28 and 24% yields, respectively, after 24 h. Accordingly, we next explored a range of chiral acid catalysts and reaction parameters using **1d** and **1e** as substrates (Supplementary Table 1). The reaction of *N*-Ts isatin **1e** in toluene at 25 °C for 50 h in the presence of 20 mol% La(OTf)₃, 22 mol% *s*-Bu-pybox **L1** and 70 mg of 4 Å molecular sieves (MS) as an additive provided chiral product **2e** in 71% yield and 98% ee (entry 1, Table 1). It was found that the catalyst loading was crucial for achieving the best enantioselectivity result

(entries 44 and 45, Supplementary Table 1). Under the same reaction conditions, product **2d** was generated from *N*-Ac isatin **1d** in a similar yield but with moderate enantioselectivity (entry 2). The ee value decreased slightly to 91% when ligand **L2** was used, as it lacked the dimethylamino moiety at the 4-position of the pyridyl unit seen in **L1** (entry 3). Other lanthanide metal triflate complexes, i.e., Ce(OTf)₃ and Nd(OTf)₃, also afforded **2e** with excellent enantioselectivity (entries 4–5). The addition of 1.0 mol% [Acr⁺–Mes]ClO₄ did not improve the reaction rate or yield (entry 6). The control experiments confirmed that the Lewis acid, visible light and an oxygen-free environment are all required for the transformation (entries 7–9). Although the use of toluene as a solvent would facilitate recycling, improving the industrial applicability of this methodology, a solvent screening was still performed to evaluate the reaction of toluene derivatives, especially solid variants (Supplementary Table 2). As a result, **2e** was obtained in 57% yield and 98% ee when using chlorobenzene (PhCl) as the solvent (entry 10).

**Substrate scope.** The generality of this direct enantioselective coupling protocol was initially explored under the optimized conditions depicted in Table 1, entry 10 (Fig. 3). We evaluated toluene derivatives with a variety of substituents on the aryl ring in the reaction with isatin **1e**. The transformations proceeded smoothly and provided products **2e-2s** in 43 to 80% yields with 83 to 98% ee within 48–60 h. For xylene and its derivatives (**2f-l**),

**Table 1 Optimization of reaction conditions**

| Entry | Variation from the standard conditions | Yield (%)[a] | ee (%)[b] |
|---|---|---|---|
| 1 | None | 71 | 98 |
| 2 | **1d** instead of **1e** | 69[c] | 70 |
| 3 | **L2** instead of **L1** | 67 | 91 |
| 4 | Ce(OTf)$_3$ instead of La(OTf)$_3$ | 67 | 95 |
| 5 | Nd(OTf)$_3$ instead of La(OTf)$_3$ | 56 | 95 |
| 6 | with 1.0 mol% [Acr$^+$-Mes]ClO$_4$ | 61 | 95 |
| 7 | No La(OTf)$_3$ | 26[d] | 13 |
| 8 | No light | 0[e] | N.A.[f] |
| 9 | Under air | 0[g] | N.A. |
| 10 | Toluene (50 equiv) in PhCl (2.0 mL) | 57 | 98 |

The reaction was performed on a 0.05 mmol scale in 1.0 mL of toluene
[a]Yield of isolated product
[b]Determined by HPLC analysis on a chiral stationary phase
[c]**2d** was obtained
[d]>95% conversion
[e]No reaction
[f]N.A. = not available
[g]Benzaldehyde was obtained

only one methyl group was functionalized. Derivatives such as 4-fluorotoluene, 4-chlorotoluene and 4-bromotoluene, which have electron-deficient substituents, provided products **2m-o** in moderate yields with 92 to 97% ee. 4-Nitrotoluene and 4-(trifluoromethyl)toluene, which have stronger electron-withdrawing substituents on the aryl ring, were also evaluated, but nearly none of the desired product was obtained; instead, isatin **1a** was completely transformed to its homocoupling product, i.e., *N,N′*-ditosylisatide **3** (0% ee, >19:1 dr). Inspiringly, the reaction of toluene-$d_8$ and heteroaromatic ring-containing 2,5-dimethylfuran and 2,5-dimethylthiophene afforded chiral deuterated oxindole **2t** and products **2u-v**, respectively, in high enantioselectivities. Secondary benzylic derivatives were next tested in the developed reaction. Nonsymmetric acyclic substrates furnished products **2w-ab** with 88 to 99% ee but low diastereomeric ratios. Diphenylmethane gave **2ac** in 50% yield with 97% ee. Cyclic 9,10-dihydroanthracene, xanthene, fluorene and acenaphthene efficiently produced the corresponding oxindoles **2ad-ag**. The formation of quaternary carbons was achieved using cumene; product **2ah** was obtained in 92% ee. No regioselectivity was observed for *p*-cymene, and the two products **2ai** and **2aj** were isolated in a ratio of nearly 1:1. Distinct isatins were also tested in the reaction with toluene, and adducts **2ak-2ap** with different substitution patterns on the aromatic ring of the oxindole were generated with 92 to 98% ee. Of note, many oxindoles with complex 3-benzyl groups, such as **2i-l**, **2t-v** and **2ac-ah**, were successfully synthesized, whether in a racemic or in an optically pure manner.

We next performed these transformations neat in toluene or one of its derivatives (see the standard reaction conditions in Table 1). Except for several solid (**2j**, **2k**, **2q**, **2ab** and **2ad-ag**) and expensive (**2 v** and **2 y**) feedstocks, toluene and all of its previously tested derivatives were shown to react efficiently with *N*-Ts isatins and achieve similar excellent enantioselectivities with the exception of **2 l**, **2p** and **2 u** (see the data in parentheses in Fig. 3). Remarkably increased yields were obtained for many

substrates. The reaction of **1e** with toluene on a 1.0 mmol scale provided product **2e** in a similar yield and enantioselectivity after 72 h (footnote a). Notably, the poor to moderate yields in all cases are due to the competition with isatin homocoupling and other resulting side transformations, given the observation of *N,N′*-ditosylisatide **3** and some unknown byproducts.

**Mechanistic studies.** Based on the structures of the products and the control experiments (Table 1), it is reasonable that the transformation occurs via sp$^3$ C–H abstraction followed by a radical coupling (Fig. 2). To elucidate the mechanism, UV-vis absorption spectroscopy analyses were performed for the reaction of isatin **1e** with toluene;[35] the results revealed that no electron donor–acceptor (EDA) complex[49] was present in the mixture. It was also found that the $E_{red\ 1/2}^\star$ of **1e** could be increased to +2.0 V in the presence of La(OTf)$_3$ (Supplementary Note 1). As mentioned by Melchiorre[12], even the photoexcited iminium ions ($E_{red\ 1/2}^\star \approx +2.4$ V), which have higher excited-state reduction potentials than **1e**, were unable to oxidize the toluene derivatives that contained electron-deficient substituents, as these compounds feature higher oxidation potentials than toluene[50]. However, our reaction system did successfully generate adducts **2m-2o** from 4-F-, 4-Cl- and 4-Br-toluene (Fig. 3). We also performed the reaction of **1e** with THF under the established conditions (Fig. 4a). Although the BDE of an α-sp$^3$ C–H bond in THF ($92.0 \pm 1.0$ kcal × mol$^{-1}$)[51] is higher than that of the sp$^3$ C–H bonds of toluene, the reaction provided coupling adduct **4** in 21% yield. These results suggest that an HAT process between the isatins and toluene and its derivatives does generate a benzylic radical species. We next attempted the reaction of 4-benzyl Hantzsch ester (**5**)[52] with **1e**, and adduct **2e** was obtained in 89% yield with 87% ee (Fig. 4b). This SET-enabled radical coupling provided the same high enantioselectivity as the HAT reaction, suggesting that using chiral La-complex as a Lewis acid would provide similar stereocontrol in both catalytic systems. Therefore,

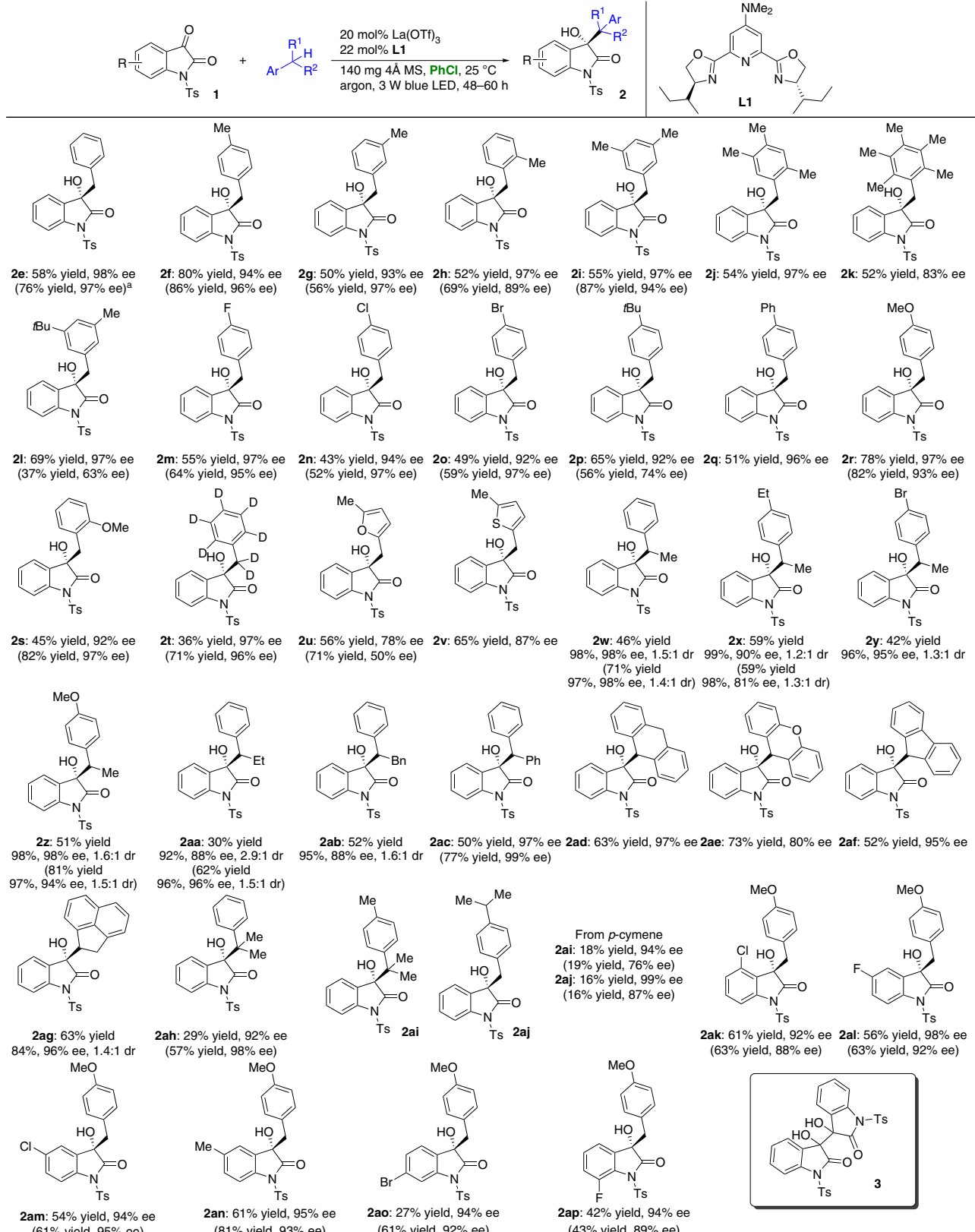

**Fig. 3** Enantioselective coupling of *N*-tosyl isatins with toluene and its derivatives. Reaction conditions: **1** (0.1 mmol), toluene or its derivatives (5.0 mmol), PhCl (4.0 mL), 25 °C. The data in parentheses were obtained under neat conditions [**1** (0.1 mmol), toluene or its derivatives (4.0 mL), 25 °C]. Yields were determined from the material isolated after chromatographic purification. Enantiomeric excesses were determined by HPLC analysis on a chiral stationary phase. ªOn a 1.0 mmol scale, 72 h, yield of **2e** = 69%, ee of **2e** = 98%

**Fig. 4** Experimental studies providing insight into the mechanism of the reaction. **a** The transformation of **1e** with THF. **b** The transformation of **1e** with 4-benzyl Hantzsch ester **5**. **c** Kinetic isotopic effect experiments. **d** The transformation of *N,N′*-ditosylisatide **3** with toluene

the chiral La-complex should interact with the O atom of ketyl species **III** or dioxindolyl radical **IV** and the O atom of the amide simultaneously. For this reason and given the lack of regioselectivity with *p*-cymene[53], we speculate that for toluene derivatives with lower oxidation potentials than toluene, the generation of the benzyl radicals can occur via HAT, HAT and SET, or SET depending on the difference between their required energies.

A competition deuterium kinetic isotope effect (KIE) study, using a mixture of toluene and toluene-$d_8$, was then performed, and the results revealed a KIE of 3.8 (Fig. 4c). On the other hand, independent measurement of the reaction rate of these two substrates indicated a remarkable difference in rate (KIE = 3.77) when both transformations proceeded within 6 h (Fig. 4c and Supplementary Fig. 109). These values suggest that the C−H bond cleavage event is involved in the first irreversible step of the catalytic cycle and therefore does affect the rate of the transformation. Notably, although **1e** was almost consumed within 30 h, the best yield of product **2e** was obtained when the reaction was prolonged to 60 h (Supplementary Fig. 109). Meanwhile, the highest yield of **2t** was achieved after the same reaction time. The

observations reflect the fact that the overall efficiency of the transformation also depends on other competing processes. Given the demonstrated capability of isatide to produce a dioxindolyl radical via homolysis[42,44], we speculated that the readily generated homocoupling byproduct *N,N′*-ditosylisatide **3**[42,44] might reversibly participate in the transformations to provide adducts **2**. In this context, the reaction of *N,N′*-ditosylisatide **3** with toluene was evaluated under the simulated reaction conditions, and as a result, product **2e** was obtained in 73% yield with 90% ee (Fig. 4d). The formation of **2e** indicates that some proportionation[42,44] of dioxindolyl radical **IV** derived from **3** occurred during the course of the reaction. Importantly, the yields of **2** could not be further improved by prolonging the reaction time after 60 h, likely due to the decreased concentration of **3** or **IV** impeding the ability to undergo homolysis or proportionation.

**Extension of the method to acenaphthoquinone**. Encouraged by these successes, we extended this method to acenaphthoquinone (**6**) to directly access another series of potentially bioactive chiral

**Fig. 5** Enantioselective coupling of acenaphthoquinone with toluene and its derivatives. 0.1 mmol scale at −5 °C under an argon atmosphere. Yields were determined from the material isolated after chromatographic purification. Enantiomeric excesses were determined by HPLC analysis on a chiral stationary phase

tertiary alcohols[54,55]. The initial study on the neat reaction of **6** with toluene under standard conditions provided product **7a** in 71% yield with 55% ee. Attempts to modify the reaction parameters could not produce better results. A further examination of the conditions was thus performed by investigating the catalytic system (Supplementary Table 3). Pleasingly, the reaction of **6** neat in toluene with 20 mol% 8H-SPINO CPA (**CPA-1**) at −5 °C with 80 mg of 4 Å MS as an additive furnished product **7a** in 51% yield with 90% ee within 60 h (Fig. 5). Five toluene derivatives were then evaluated, and they produced adducts **7b-f** in moderate yields with good enantioselectivities under slightly modified reaction conditions (Supplementary Methods).

## Discussion

In summary, we have developed a chiral acid-catalysed enantioselective direct C(sp$^3$)−H functionalization of toluene and its derivatives with activated ketones under irradiation with visible light. In the presence of a chiral Lewis acid catalyst, a wide range of biologically important 3-hydroxy-3-benzyl-substituted 2-oxindoles including many difficult-to-access variants were obtained from isatins in a straightforward manner in high yields and with high ee. This challenging C−H abstraction and radical coupling strategy is also compatible with acenaphthoquinone when a CPA catalyst is used, furnishing access to another array of valuable enantioenriched tertiary alcohols. Given the tremendous achievements based on the Paternò–Büchi reaction and its variants, we believe that this result will inspire the further pursuit of asymmetric scaffolds and transformations, which will provide expedient and economical approaches for synthesizing numerous useful chiral compounds from diverse ketones and abundant chemical feedstocks.

## Methods

**General information**. For the NMR spectra of compounds in this manuscript, see Supplementary Figs. 1−46. For the HPLC spectra of compounds in this manuscript, see Supplementary Figs. 47−92. For details of the optimization of reaction conditions, see Supplementary Note 1. For details of the mechanistic studies, see Supplementary Note 2. For the determination of the absolute configuration of products, see Supplementary Note 3. For the Gaussian data, see Supplementary Data. For general information, general experimental procedures and analytic data of compounds synthesized, see Supplementary Methods. For references on DFT caculations, see Supplementary Reference.

**Preparation of 2 with PhCl as the solvent**. First, **1** (0.1 mmol, 1.0 equiv), **L1** (0.022 mmol, 0.22 equiv), La(OTf)$_3$ (0.02 mmol, 0.2 equiv) and 4 Å MS (140 mg) were added to a 25 mL Schlenk tube. Subsequently, toluene or one of its derivatives (5.0 mmol, 50.0 equiv) and PhCl (4.0 mL) were sequentially added and degassed three times by the freeze–pump–thaw method. The reaction mixture was stirred under an argon atmosphere at 30 °C (the temperature was maintained in an incubator) in the dark for 2 h and then irradiated with a 3 W blue LED (λ = 450–455 nm) from a distance of 3.0 cm for another 48–60 h at 25 °C. The reaction mixture was directly loaded onto a short silica gel column, followed by gradient elution with petroleum ether/ethyl acetate (20/1–5/1 ratio). Removing the solvent in vacuo afforded products **2e-an**.

**Preparation of 2 in neat toluene and its derivatives**. First, **1** (0.1 mmol, 1.0 equiv), **L1** (0.022 mmol, 0.22 equiv), La(OTf)$_3$ (0.02 mmol, 0.2 equiv) and 4 Å MS (140 mg) were added to a 25 mL Schlenk tube. Subsequently, toluene or one of its derivatives (4.0 mL) was sequentially added and degassed three times by the freeze–pump–thaw method. The reaction mixture was stirred under an argon atmosphere at 30 °C (the temperature was maintained in an incubator) in the dark for 2 h and then irradiated with a 3 W blue LED (λ = 450−455 nm) from a 3.0 cm distance for another 48−60 h at 25 °C. The reaction mixture was directly loaded onto a short silica gel column, followed by gradient elution with petroleum ether/ethyl acetate (20/1-5/1 ratio). Removing the solvent in vacuo afforded products **2e-i**, **2l-p**, **2r-2u**, **2w-aa**, **2ac** and **2ag-ap**.

**Preparation of 7a**. First, **6** (0.1 mmol, 1.0 equiv), **CPA-1** (0.02 mmol, 0.20 equiv) and 4 Å MS (80 mg) were added to a 25 mL Schlenk tube. Subsequently, toluene (saturated aqueous solution) (2.0 mL) was added and degassed three times by the freeze–pump–thaw method. The reaction mixture was stirred under an argon atmosphere at −5 °C (the temperature was maintained in an incubator) for 30 min in the dark and then irradiated with 2*3 W blue LEDs (λ = 450−455 nm) from a distance of 6.0 cm for another 60 h at −5 °C. The reaction mixture was directly loaded onto a short silica gel column, followed by elution with a gradient of petroleum ether/dichloromethane (1/1 ratio) to petroleum ether/ethyl acetate (5/1 ratio). Removing the solvent in vacuo afforded product **7a**.

## Data availability

The X-ray crystallographic coordinates for structures that support the findings of this study have been deposited at the Cambridge Crystallographic Data Centre (CCDC) with the accession codes CCDC 1896394 (**3**), 1877077 (**7b**), 1877066 (**8**). These data can be obtained free of charge from The Cambridge Crystallographic Data Centre via www.ccdc.cam.ac.uk/data_request/cif. The authors declare that all other data supporting the findings of this study are available within the article and Supplementary Information files and are also available from the corresponding author upon reasonable request.

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

## Acknowledgements

Grants from the NSFC (21672052) and Henan University are gratefully acknowledged. We also appreciate Prof. Qiang Liu (Lanzhou University) for providing constructive discussions.

## Author contributions

Z.J. conceived and designed the experiments. F.L., D.T. and Y.F. performed the experiments. R.L. performed DFT calculations on the triplet-state energies of compounds. G.L. performed DFT calculations on BDEs. Y.Y., B.Q., X.Z. and Z.X. helped isolate compounds and analyse the data. Z.J. wrote the paper. F.L., D.T. and Y.F. contributed equally to this work. All authors discussed the results and commented on the manuscript.

## Additional information

**Competing interests:** The authors declare no competing interests.

