## [Peer Review File · Nature Communications]

Reviewers' comments:

Reviewer #1 (Remarks to the Author):

The manuscript describes two photochemical catalytic systems for the stereoselective C(sp³)-H functionalization of feedstock chemicals, including toluene and derivatives, with photoactive ketones. Mechanistically, the ketone absorbs visible-light to reach a triplet excited state and promote a hydrogen atom abstraction (HAT) from the benzylic position of toluene scaffolds, thus furnishing the open-shell benzylic radical. The aggregation of a chiral acid catalyst to the generated ketyl radical (IV in Figure 2, better said dioxindolyl radical, see below) enables the stereoselective coupling of the two open-shell intermediates, ultimately delivering the corresponding alcohol products with good to excellent level of enantioinduction. This strategy has been applied to the benzylation of isatins. A second approach takes advantage of the photoactivity of acenaphthoquinone, and the stereoinduction is imparted by a Brønsted chiral phosphoric acid. The scope of the transformation is broad with respect to the toluene derivatives and the alcohol products are obtained in high enantiomeric excess in the most of the cases.

Synthetically, this chemistry is interesting, although the same products can be accessed by the enantioselective α -oxygenation of 3-benzyl-substituted 2-oxindoles. But developing photochemical catalytic asymmetric processes with such a high degree of stereoinduction is far from easy, and this study has found an effective solution to this problem. From this angle, the chemistry detailed in this contribution is interesting and of high impact, and could deserve publication within this journal. However, the inattentive/superficial manner in which the manuscript is written and the results are presented (especially concerning the mechanistic studies) detracts from the scientific appeal of the paper. A thorough revision of the manuscript is strongly encouraged in order to secure its publication in Nature Communications.

Some comments follow.

The major concerns are about the way the manuscript is written and the mechanistic studies. This manuscript is difficult to follow, and many concepts are not clear or explained in a superficial manner. Language polishing is strongly recommended before of resubmission.

As for the mechanistic studies, different aspects are unconvincing. The main concern is about the proposed radical coupling mechanism, which requires the persistency of one of the two radicals involved (according to the persistent radical effect). Unfortunately, the authors have not discussed this central point at all. It is granted that benzyl radicals are not persistent, but the structure of intermediates III and IV hints to this possibility and should be better discussed. For example, intermediate IV is a merostabilized carbon free radical, as it is stabilized by dipolar resonance structure (captodative effect: the radical center lies between the electron-donating hydroxyl substituent and the electron-withdrawing carbamido substituent). Similar radicals have been reported in the literature, and named dioxindolyl radicals (*J. Am. Chem. Soc.* 1980, 102, 2345). Also intermediate III has been reported (*J. Am. Chem. Soc.* 1970, 92, 2762). The authors should better discuss the nature of the proposed radical intermediates and contextualize them on the basis of previous literature reports, and discuss for example if the dimer of IV (isatide) has been observed under the reaction conditions.

Along the same line, in Figure 2 the authors have inferred the value of the bond dissociation energy (BDE) for intermediate IV based on the BDE value for iso-propanol. I am sceptical about the validity of this approximation, due to the structural difference with the isatin core.

In Figure 3, example 2s: the possibility to use toluene-d-8 should be better discussed in terms of mechanistic implications: the evaluation of a kinetic isotope effect could provide valuable information about the deprotonation/HAT step within the photochemical process and should be adequately discussed.

Other minor aspects are listed below:

- Line 25-26: the wording should be "difficult-to-access".

- Line 50: Paternò and Cheffi are both misspelled.
- Figure 2: the excited-state potentials for isatins 1a, 1b and 1e do not match the one reported in the Supporting Information.
- Line 67 and reference 34: the literature includes more stereoselective α -oxidation methods, delivering 3-hydroxy-3-benzyl-substituted 2-oxindoles, other than the one reported by the authors in ref. 34 (Chem. Eur. J. 2012, 18, 8916–8920; J. Org. Chem. 2015, 80, 12686–12696; Tetrahedron Lett. 2018, 59, 2412–2417). For completeness, these should be included in the manuscript together with other methods affording analogous scaffolds (Org. Lett. 2016, 18, 1358–1361; Org. Lett. 2018, 20, 6183–6187).
- Line 95: the oxidation, under aerobic conditions, of toluene likely delivers benzaldehyde and not acetophenone.
- Line 98: the reaction “delivered a complex crude mixture” is preferable to the wording “messy”.
- Line 114: the chemical name of PhCl is “chlorobenzene” and not “benzyl chloride”.
- Figure 3: The amount (equivalents) of toluene derivative employed should be provided either in the reaction diagram or in the figure caption.
- Line 147: the wording “C(sp³)-H functionalization/abstraction” would be preferable to “sp³ C-H activation”, since no transition-metal insertion within the C-H bond is occurring. This is repeated in line 184.
- Line 151: iminium ions are not “photogenerated” but “photoexcited”.
- Figure 4, example 6e: the structure of the product has likely an exceeding carbon at the benzylic position.
- Line 180: the wording “directed C(sp³)-H functionalization” would be preferable.
- In the Supporting Information, section 4, page S16: the cyclic voltammetry diagrams look of difficult interpretation, presumably due to the very high scan rate at which they have been recorded. Repeating and comparing the analyses at a lower scan rate would enable a more precise/reliable potential values measurement.

To sum up, the chemistry detailed in this manuscript is highly interesting, but the manuscript does not reach the level of accuracy required for publication in Nature Communications.

Reviewer #2 (Remarks to the Author):

This manuscript by Zhiyong Jiang and co-workers describes the enantioselective catalytic functionalization of toluene derivatives. Two different classes of dicarbonyl substrates are employed, 2-oxindoles and acenaphthoquinones. Given their different reactivity, different types of catalysts are employed. With 2-oxindoles a La(OTf)₃/pybox-based catalyst was employed while with acenaphthoquinone a chiral phosphoric acid was utilized. In both cases, the chemistry capitalizes on the photochemistry of the substrate, and no photocatalysts are necessary. The approach does follow in the footsteps of Melchiorre’s work (JACS 2018, 140, 8439) but that does not detract from the present work because the chemistry is unrelated.

“Isatin 1a was found to work but the reaction was messy, and no reaction was observed for 1b or 1c” How was it determined that 1a worked? NMR?

The resolution of the schemes is so low that some parts are difficult to read. Please fix.

“...when using benzyl chloride (PhCl) as the solvent (entry 10).” Chlorobenzene not benzyl chloride.

The scope study in Figure 3 indicates that the reaction is quite broad and highly enantioselective. A variety of toluenes possessing multiple methyl groups and electron withdrawing groups seemed to work very well. Secondary benzylic C–H’s are also viable in the enantioselective reaction, giving good enantioselectivities (but poor diastereoselectivities). The only type of substrates that are underrepresented are heterocyclic derivatives, which are probably the most important. Please

either add some or explain which ones were tried but failed. There are not many electron withdrawing groups. Please state which ones failed to undergo reaction so the reader can know the limitations of the method.

Why is such a high loading of the chiral Lewis acid needed? Some comment should be made about examining different loadings in Table S1.

The SI is clearly written, HPLC traces indicate good separation of the enantiomers, and the NMR spectra are clean. Are any of these compounds known? This must be stated. Photographs of the reaction setups should be included in the SI with details of the distance from the light to the reactor and the orientation.

Overall, this is very nice chemistry and I believe it merits publication in a top tier journal after the modifications outlined above are made. Toluenes are useful feedstocks and reactions that can use them, even to give racemic or achiral compounds, are important. Enantioselective reactions such as those shown here are indeed impressive.

Reviewer #3 (Remarks to the Author):

Jiang and coworkers reported a chiral acid-catalysed enantioselective direct sp^3 -functionalization of toluene and its derivatives with ketones under the irradiation of visible light. By utilizing combination of $La(OTf)_3$ and *s*-Bu-pybox as the chiral catalyst, a wide range of 3-hydroxy-3-benzyl-substituted 2-oxindoles involving many difficult-to-synthesize variants were obtained from isatins in a straightforward manner in good to excellent yields and enantioselectivities. While by using a chiral phosphoric acid catalyst, acenaphthoquinone was also compatible yielding coupling adducts as well. Without using external agent, the reaction was economical and environmentally friendly. Compared with tradition methods, this work opens a door for the asymmetric synthesis of inert $C(sp^3)$ -H functionalization. This paper would attract chemists especially those dealing with C-H functionalization and visible light-driven photoredox catalysis reactions. Therefore, the referee would be like to receive this manuscript after minor revision.

Questions:

1. The authors discuss electron-donating and electron-neutral substituents of toluene derivatives, what about electron-drawing substituents of toluene?
2. Some toluene derivatives gave middle yield of the adducts, what happened to these reactions?
3. Correct the sentence, page no. 9, first paragraph and last line: "..... complex 3-benzyl groups such as 2i-2l, 2s and 2aa-2af were first time synthesized....".
4. Some related papers about the important of 3-hydroxy-3-benzyl-substituted 2-oxindoles have recently appeared and they should be included in the reference list: such as Asian J. Org. Chem. 2018, 7, 337-340; Asian J. Org. Chem. 2018, 7, 11429-11438 and so on.

Point to Point Response to Reviewers' Comments

Please take note that all the descriptive, positive comments of the reviews are omitted, and only reviewer's comments expressing their concerns/suggestions are listed below, which are followed by our response.

Reviewer 1' comments and our responses

1) Language polishing is strongly recommended before of resubmission.

Response: We have revised the logic of the manuscript carefully. A proofreading was performed for the revised manuscript before resubmission.

2) As for the mechanistic studies, different aspects or unconvincing. The main concern is about the proposed radical coupling mechanism, which requires the persistency of one of the two radicals involved (according to the persistent radical effect). Unfortunately, the authors have not discussed this central point at all. It is granted that benzyl radicals are not persistent, but the structure of intermediates III and IV hints to this possibility and should be better discussed. For example, intermediate IV is a merostabilized carbon free radical, as it is stabilized by dipolar resonance structure (captodative effect: the radical center lies between the electron-donating

hydroxyl substituent and the electron-withdrawing carbamido substituent). Similar radicals have been reported in the literature, and named dioxindolyl radicals (J. Am. Chem. Soc. 1980, 102, 2345). Also intermediate III has been reported (J. Am. Chem. Soc. 1970, 92, 2762). The authors should better discuss the nature of the proposed radical intermediates and contextualize them on the basis of previous literature reports, and discuss for example if the dimer of IV (isatide) has been observed under the reaction conditions.

Response: Thanks for the help comments. We have carefully read the two literatures and discuss these knowledge points in the manuscript, including radical anion **II**, dioxindolyl radical **IV**, intermediate **V** as the merostabilized carbon radical like **IV**, the captodative effect and the persistent radical effect. Please read page 6 in the revised manuscript. Moreover, both literatures and a literature on the persistent radical effect have been added as ref. 43–45 in the revised manuscript.

Yes, isatide was found in our reaction system. In the revised manuscript, this side product was mentioned in Fig. 3 on page 9, the second paragraph on page 10, Fig. 4 on page 12 and page 13. Isatide was found to furnish a dioxindolyl radical **IV** via homolysis, which can further experience proportionation to form the starting substrate, i.e., isatin. The experiment as shown in Fig. 4d and Supplementary Figure 109 could support the deduction. Indeed, these transformations were also described in the literatures recommended by the referee.

3) Along the same line, in Figure 2 the authors have inferred the value of the bond dissociation energy (BDE) for intermediate IV based on the BDE value for iso-propanol. I am sceptical about the validity of this approximation, due to the structural difference with the isatin core.

Response: DFT calculations on the BDE energies of these intermediates were performed. The results could be found in Supplemental Table 7. Yes, they are totally different with that of iso-propanol. Many thanks!

4) In Figure 3, example 2s: the possibility to use toluene-*d*-8 should be better discussed in terms of mechanistic implications: the evaluation of a kinetic isotope effect could provide valuable information about the deprotonation/HAT step within the photochemical process and should be adequately discussed.

Response: A competition deuterium kinetic isotope effect (KIE) study, using a mixture of toluene and toluene-*d*₈, was performed, and the results revealed a KIE of 3.8 (Fig. 4c). However, independent measurement of the reaction rate of these two substrates indicated that C–H cleavage does not determine the reaction rate in terms of achieving the best yield, while a remarkable difference in rate (KIE = 3.77) was observed before 6 h (Fig. 4c and Supplementary Figure 109). Since *N,N'*-ditosylisatin **3** can undergo homolysis and then proportionation to furnish the starting substrate, i.e., isatin, the reaction time has to be prolonged to achieve the best yield.

All results and comments could be found in Fig. 4 on page 12, on page 13 and in Supplementary Figure 109.

5) Other minor aspects are listed below:

- Line 25-26: the wording should be “difficult-to-access” .
- Line 50: Paternò and Cheffi are both misspelled.
- Figure 2: the excited-state potentials for isatins 1a, 1b and 1e do not match the one reported in the Supporting Information.

Response: These issues have been addressed.

- Line 67 and reference 34: the literature includes more stereoselective α -oxidation methods, delivering 3-hydroxy-3-benzyl-substituted 2-oxindoles, other than the one reported by the authors in ref. 34 (Chem. Eur. J. 2012, 18, 8916 – 8920; J. Org. Chem. 2015, 80, 12686 – 12696; Tetrahedron Lett. 2018, 59, 2412 – 2417). For completeness, these should be included in the manuscript together with other methods affording analogous scaffolds (Org. Lett. 2016, 18, 1358 – 1361; Org. Lett. 2018, 20, 6183 – 6187).

Response: The literatures have been cited as ref 35–39.

- Line 95: the oxidation, under aerobic conditions, of toluene likely delivers benzaldehyde and not acetophenone.
- Line 98: the reaction “delivered a complex crude mixture” is preferable to the wording “messy” .
- Line 114: the chemical name of PhCl is “chlorobenzene” and not “benzyl chloride” .

Response: These issues have been addressed.

- Figure 3: The amount (equivalents) of toluene derivative employed should be provided either in the reaction diagram or in the figure caption.

Response: The equivalents of these starting substrates were mentioned in the reaction diagram. Please find it in Fig. 3.

- Line 147: the wording “C(sp³)-H functionalization/abstraction” would be preferable to “sp³ C-H activation” , since no transition-metal insertion within the C-H bond is occurring. This is repeated in line 184.
- Line 151: iminium ions are not “photogenerated” but “photoexcited” .
- Figure 4, example 6e: the structure of the product has likely an exceeding carbon at the benzylic position.
- Line 180: the wording “directed C(sp³)-H functionalization” would be preferable.
- In the Supporting Information, section 4, page S16: the cyclic voltammetry diagrams looks of difficult interpretation, presumably due to the very high scan rate at which they have been recorded. Repeating and comparing the analyses at a lower scan rate would enable a more precise/reliable potential values measurement.

Response: These issues have been addressed.

Reviewer 2' comments and our responses

- 1) “Isatin 1a was found to work but the reaction was messy, and no reaction was observed for 1b or 1c” How was it determined that 1a worked? NMR?

Response: Indeed, the reaction of **1a** worked in only ~10% chemical conversion determined by TLC analysis. The reaction delivered a complex crude mixture and the desired product **2a** could not be obtained. These comments have been added in *footnote c*, Supplementary Table 1

2) The resolution of the schemes is so low that some parts are difficult to read. Please fix.

Response: The issue has been addressed.

3) “...when using benzyl chloride (PhCl) as the solvent (entry 10).” Chlorobenzene not benzyl chloride.

Response: Done.

4) The scope study in Figure 3 indicates that the reaction is quite broad and highly enantioselective. A variety of toluenes possessing multiple methyl groups and electron withdrawing groups seemed to work very well. Secondary benzylic C - H's are also viable in the enantioselective reaction, giving good enantioselectivities (but poor diastereoselectivities). The only type of substrates that are underrepresented are heterocyclic derivatives, which are probably the most important. Please either add some or explain which ones were tried but failed. There are not many electron withdrawing groups. Please state which ones failed to undergo reaction so the reader can know the limitations of the method.

Response: As requested, we attempted 2,5-dimethylthiophene in the reaction. The product **2v** was obtained in 65% yield with 87% ee. We then examined 4-fluorotoluene which contains a higher electron-withdrawing group than 4-chlorotoluene and 4-bromotoluene. As shown in Fig. 3, moderate yield with excellent enantioselectivity of adduct **2m** were obtained. Certainly, 4-nitrotoluene and 4-(trifluoromethyl)toluene with a stronger electron-withdrawing substituent on the aryl ring were also evaluated, but nearly no desired product was obtained, wherein isatin **1a** was completely transformed to its homocoupling product, i.e., isatide **3**. The comments were added on page 10 in the

revised manuscript.

- 5) Why is such a high loading of the chiral Lewis acid needed? Some comment should be made about examining different loadings in Table S1.

Response: The comment 'It was found that the catalyst loading was crucial for achieving the best enantioselective result (entries 44 and 45, Supplementary Table 1)' has been added on page 8 in the revised manuscript.

- 6) Are any of these compounds known? This must be stated. Photographs of the reaction setups should be included in the SI with details of the distance from the light to the reactor and the orientation.

Response: In the manuscript, we have mentioned which types of products are undeveloped in the previous examples. Actually, given *N*-Ts as the substituent, all products are unknown. The distance from the light to the reactor has been described in the 'General experimental procedures' wherein the photographs of the reaction setups were added (See Supplementary Note 2).

Reviewer 3' comments and our responses

- 1) The authors discuss electron-donating and electron-neutral substituents of toluene derivatives, what about electron-drawing substituents of toluene?

Response: 4-Fluorotoluene has been tested and the results are satisfactory. No reaction was found for 4-nitrotoluene and 4-(trifluoromethyl)toluene as the starting substrates. The results and the comments have been added in the revised manuscript.

- 2) Some toluene derivatives gave middle yield of the adducts, what happened to these reactions?

Response: The comments 'The poor to moderate yields in all cases are due to the generation of some unknown byproducts including isatide **3**. Even extending the reaction time, **3** still could not be fully exhausted.' have been added in the revised manuscript.

3) Correct the sentence, page no. 9, first paragraph and last line: “.....
complex 3-benzyl groups such as 2i-2l, 2s and 2aa-2af were first time
synthesized...” .

Response: Done. Thanks!

4) Some related papers about the important of 3-hydroxy-3-benzyl-substituted
2-oxindoles have recently appeared and they should be included in the reference list:
such as Asian J. Org. Chem. 2018, 7, 337 - 340; Asian J. Org. Chem. 2018, 7,
11429-11438 and so on.

Response: The two literatures have been added as ref. 40–41 in the revised manuscript.

Sincerely,

Zhiyong

Reviewers' comments:

Reviewer #1 (Remarks to the Author):

In the revised version of their manuscript, the authors have addressed the major mechanistic concerns identified in the previous round of evaluation, thus providing a more sounding and coherent mechanistic picture of the process. Frankly, I am a little concerned that the formation of the isatide byproducts, which is now extensively discussed, was completely overlooked in the previous manuscript. This intermediate is formed during the reaction and it is actively involved in the mechanism, acting as a sort of reservoir for the formation of the dioxindolyl radical, as demonstrated by the new results discussed in Figure 4d of the revised manuscript. The fact that this reactivity behavior and the large body of literature discussing the reactivity of dioxindolyl radicals went completely unnoticed (or unmentioned) in the previous manuscript indicates the poor attention the authors put on the mechanistic details. This lack of details transpire out of the entire manuscript, which remains difficult to follow, since many concepts are still explained in a superficial and confusing manner. Despite the authors' claim that language polishing has been done and the logic of the manuscript improved, extensive rewording will be needed to bring this manuscript to an acceptable level for publication within the journal. Still, we are here to mainly judge the science, and I believe that the reported results are highly interesting and worth of publication in Nature Communication.

There are still aspects that need to be carefully addressed before of publication, which are listed below:

This manuscript is difficult to follow. Language polishing and a more logical mechanistic discussion are strongly recommended before of publication.

In Figure 2, it is not clear how the chiral acid interacts with the reactive radical intermediate – intermediate V should have a proton connecting the phosphate anion and the carbonyl oxygen of the substrate, thus giving a stabilized dioxindolyl radical, while the proposed interaction in intermediate VI is unclear: again, a proton is missing on the catalyst scaffold.

- In line 42-43: the statement “the formation of new chiral C(sp³)- C(sp³) bonds” is scientifically meaningless: chirality is a property of a whole molecule and a bond cannot be defined as chiral. This must be addressed.
- The sentence within lines 43-45 is rather confusing. The authors presumably meant that the use of a radical-based strategy offers improved regioselectivity with respect to transition-metal catalysed C-H activation procedures, especially when it comes to the functionalization of toluene derivatives (insertion into C(sp³)-H vs C(sp²)-H bond). This is due to the greater stability of a benzylic radical compared to an aryl one. If this is the concept the authors aim to convey, the sentence should be clarified and reformulated.
- In line 51: the definition “iminium catalysis” in this context is misleading, since it could refer to the iminium mode of activation in the ground state domain. The wording “excited-state iminium ion catalysis” would be more appropriate.
- In line 58: the ketones are photoexcited, not photoinduced.
- In Figure 3: the reaction diagram reports that 50 equivalents of toluene derivatives, compared to the isatin substrate, are employed in each entry. Is this stoichiometry referred solely to the neat reactions or is it applied also to the experiments using chlorobenzene as solvent? Please, clarify.
- Line 84: “which is a merostabilized carbon radical similar IV due to the captodative effect” – this sentence is unclear and should be changed; what does exactly mean “carbon radical similar IV”?
- In line 138: the wording “ees” is incorrect, as ee does not have a plural form. Please, correct throughout the manuscript.
- In line 144: the letter D to identify the deuterated form of toluene should be written lowercase and in italics (toluene-d₈).
- In line 185 and 186: the reaction component providing stereocontrol by coordination with the isatin core is not La(OTf)₃ itself but rather a chiral La-complex, since ligated with the chiral ligand

L1.

- The sentence within lines 192-195 is imprecise. The measured KIE = 3.77 clearly indicates that the C-H cleavage event is involved in the first irreversible step of the catalytic cycle and, therefore, it does affect the rate of the transformation. Claiming that "does not determine the reaction rate in terms of achieving the best yield" reflects the fact that the whole efficiency of the process depends also on other competing processes, including the homocoupling of the dioxindolyl radical to generate 3 and the subsequent homolytic cleavage of the latter to afford the reaction product 2. For this reason, longer reaction times are required to achieve optimal efficiency.
- In line 210: "neat" is misspelled.
- In line 218: the title of the subchapter is presumably "Conclusions" rather than "Discussion".
- In line 255 and 262: the synthetic procedure likely reports the preparation of compound 7a.

To sum up, this is a scientifically interesting manuscript, which unfortunately has been written in a superficial and confusing manner while the experimentation has been conducted with little attention to mechanistic details. Still, I am of the opinion that the scientific results deserve publication in Nature Communications.

Reviewer #2 (Remarks to the Author):

Since this is the second review of this manuscript, I will skip the usual introduction. Overall, the manuscript has been improved by the authors. There are still some issues with the new material. In particular, the KIE experiments are not outlined in a way that I can follow. I do not understand what they did. This needs to be revised.

In the abstract and conclusions, the using the work "ketones" overgeneralizes the chemistry. Two specific classes of strongly highly activated ketones are used, isatins and a 1,2-dione.

Has the stereochemistry of 3 been confirmed? What is the dr?

The authors write: "A competition deuterium kinetic isotope effect (KIE) study, using a mixture of toluene and toluene-d8, was then performed, and the results revealed a KIE of 3.8 (Fig. 4c). However, independent measurement of the reaction rate of these two substrates indicated that C-H cleavage does not determine the reaction rate in terms of achieving the best yield, while a remarkable difference in rate (KIE = 3.77) was observed before 6 h (Fig. 4c and Supplementary Figure 109)." This is not clear at all to me. Please rewrite it.

Also from KIE experiments "The reaction was conducted with 1b as limiting reagent." Do the authors mean 1e?

They also write on page 119 of the SI "The reaction mixture was stirred under an argon atmosphere at 30 oC (the temperature was maintained in an incubator) in dark for 2 h, then irradiated by a 3 W blue LED ($\lambda = 450-455$ nm) from a 3.0 cm distance for another hours at 25 oC."

It is not clear to me how they even did the experiment.

Reviewer #3 (Remarks to the Author):

It is OK in my side, publish as it is.

Point to Point Response to Reviewers' Comments

Please take note that all the descriptive, positive comments of the reviews are omitted, and only reviewer's comments expressing their concerns/suggestions are listed below, which are followed by our response.

Reviewer 1' comments and our responses

- 1) In Figure 2, it is not clear how the chiral acid interacts with the reactive radical intermediate - intermediate V should have a proton connecting the phosphate anion and the carbonyl oxygen of the substrate, thus giving a stabilized dioxindolyl radical, while the proposed interaction in intermediate VI is unclear: again, a proton is missing on the catalyst scaffold.

Response: Thanks! Actually, in this design plan, the chiral acid is a general term that includes chiral Lewis acid and chiral Brønsted acid. In this context, 'A' refers to 'H' or 'M'.

- 2) In line 42-43: the statement “the formation of new chiral C(sp³)-C(sp³) bonds” is scientifically meaningless: chirality is a property of a whole molecule and a bond

cannot be defined as chiral. This must be addressed.

Response: The word 'chiral' has been deleted.

- 3) The sentence within lines 43-45 is rather confusing. The authors presumably meant that the use of a radical-based strategy offers improved regioselectivity with respect to transition-metal catalysed C-H activation procedures, especially when it comes to the functionalization of toluene derivatives (insertion into C(sp³)-H vs C(sp²)-H bond). This is due to the greater stability of a benzylic radical compared to an aryl one. If this is the concept the authors aim to convey, the sentence should be clarified and reformulated.

Response: The sentences have been revised as: This elegant work further demonstrates that the use of a radical-based strategy offers improved regioselectivity with respect to transition-metal catalysed C-H activation procedures, especially in regard to the functionalization of toluene derivatives, by differentiating C(sp³)-H from C(sp²)-H due to the greater stability of a benzylic radical compared to that of an aryl one. Please read page 3 in the revised manuscript.

- 4) In line 51: the definition “iminium catalysis” in this context is misleading, since it could refer to the iminium mode of activation in the ground state domain. The wording “excited-state iminium ion catalysis” would be more appropriate.

Response: Thanks. It has been revised as requested.

- 5) In line 58: the ketones are photoexcited, not photoinduced.

Response: Done.

- 6) In Figure 3: the reaction diagram reports that 50 equivalents of toluene derivatives, compared to the isatin substrate, are employed in each entry. Is this stoichiometry referred solely to the neat reactions or is it applied also to the experiments using chlorobenzene as solvent? Please, clarify.

Response: We described both reaction conditions in footnote of Fig. 3.

7) Line 84: “which is a merostabilized carbon radical similar IV due to the captodative effect” – this sentence is unclear and should be changed; what does exactly mean “carbon radical similar IV” ?

Response: We have revised the sentences as: Given the strong basicity of the oxygen anion of **III**, it might capture the chiral Brønsted acid or Lewis acid catalyst to form dioxindolyl or the dioxindolyl-like radical **IV**, which is a merostabilized carbon radical due to the capto-dative substitution of the radical centre by the electron-donating oxygen atom and the electron-withdrawing amide group. Please see page 6 in the revised manuscript.

8) In line 138: the wording “ees” is incorrect, as ee does not have a plural form. Please, correct throughout the manuscript.

Response: Thanks for this helpful comment. We have addressed all this issue.

9) In line 144: the letter D to identify the deuterated form of toluene should be written lowercase and in italics (toluene-d8).

Response: Done.

10) In line 185 and 186: the reaction component providing stereocontrol by coordination with the isatin core is not La(OTf)₃ itself but rather a chiral La-complex, since ligated with the chiral ligand L₁.

Response: Thanks! We have revised La(OTf)₃ as La-complex as requested.

11) The sentence within lines 192-195 is imprecise. The measured KIE = 3.77 clearly indicates that the C-H cleavage event is involved in the first irreversible step of the catalytic cycle and, therefore, it does affect the rate of the transformation. Claiming that “does not determine the reaction rate in terms of achieving the best yield” reflects the fact that the whole efficiency of the process depends also on other competing processes, including the homocoupling of the dioxindolyl radical to generate **3** and the subsequent homolytic cleavage of the latter to afford the reaction product **2**. For this reason, longer reaction times are required to achieve optimal

efficiency.

Response: We appreciate the referee for the help guidance. The descriptions have been revised as: On the other hand, independent measurement of the reaction rate of these two substrates indicated a remarkable difference in rate (KIE = 3.77) when both transformations proceeded within 6 h (Fig. 4c and Supplementary Fig. 109). These values suggest that the C–H bond cleavage event is involved in the first irreversible step of the catalytic cycle and therefore does affect the rate of the transformation. Notably, although **1e** was almost consumed within 30 h, the best yield of product **2e** was obtained when the reaction was prolonged to 60 h (Supplementary Fig. 109). Meanwhile, the highest yield of **2t** was achieved after the same reaction time. The observations reflect the fact that the overall efficiency of the transformation also depends on other competing processes. Given the demonstrated capability of isatide to produce a dioxindolyl radical via homolysis, we speculated that the readily generated homocoupling byproduct *N,N'*-ditosylisatide **3** might reversibly participate in the transformations to provide adducts **2**. These sentences can be found on pages 13 to 14 in the revised manuscript.

12) In line 210: “neat” is misspelled.

Response: Done. Thanks!

13) In line 218: the title of the subchapter is presumably “Conclusions” rather than “Discussion” .

Response: Actually, as requested by Nat. Commun., the definition of the last section is ‘Discussion’ but not ‘Conclusions’.

14) In line 255 and 262: the synthetic procedure likely reports the preparation of compound 7a.

Response: Yes, it’s our mistake! I have revised it. Many thanks!

Reviewer 2’ comments and our responses

- 1) In the abstract and conclusions, the using the work “ketones” overgeneralizes the chemistry. Two specific classes of strongly highly activated ketones are used, isatins and a 1,2-dione.

Response: In all cases, ‘ketones’ has been revised as ‘activated ketones’.

- 2) Has the stereochemistry of **3** been confirmed? What is the dr?

Response: Dr is >19:1 and this result has been added on page 10 in the revised manuscript.

- 3) The authors write: “A competition deuterium kinetic isotope effect (KIE) study, using a mixture of toluene and toluene-*d*₈, was then performed, and the results revealed a KIE of 3.8 (Fig. 4c). However, independent measurement of the reaction rate of these two substrates indicated that C–H cleavage does not determine the reaction rate in terms of achieving the best yield, while a remarkable difference in rate (KIE = 3.77) was observed before 6 h (Fig. 4c and Supplementary Figure 109).” This is not clear at all to me. Please rewrite it.

Response: Done. Please read the revised sentences on pages 13 to 14 in the revised manuscript.

- 4) Also from KIE experiments “The reaction was conducted with **1b** as limiting reagent.” Do the authors mean **1e**?

Response: Thanks the referee for carefulness. Yes, it is **1e** but not **1b**.

- 5) They also write on page 119 of the SI “The reaction mixture was stirred under an argon atmosphere at 30 °C (the temperature was maintained in an incubator) in dark for 2 h, then irradiated by a 3 W blue LED ($\lambda = 450-455$ nm) from a 3.0 cm distance for another hours at 25 °C.” It is not clear to me how they even did the experiment.

Response: Thanks again for carefulness. It’s our mistake and 30 °C has been revised as

25 °C.

Sincerely,

Zhiyong